# Deep neuromuscular block for minimally invasive lung surgery: a protocol for a systematic review with meta-analysis and trial sequential analysis

Jianqiao Zheng ,[1] Li Du,[2] Xiaoqian Deng,[1] Lu Zhang,[1] Jia Wang,[1] Guo Chen[1]

[1]Department of Anesthesiology, Sichuan University West China Hospital, Chengdu, Sichuan, China
[2]Department of Anesthesiology, Sichuan Cancer Hospital & Institute, Sichuan Cancer Center, School of Medicine, University of Electronic Science and Technology of China, Chengdu, Sichuan, China

**Correspondence to**
Professor Guo Chen;
Anesthesiology_SCU@163.com

## ABSTRACT

**Introduction** Minimally invasive lung surgery (MILS) gradually became the primary surgical therapy for lung cancer, which remains the leading cause of cancer death. Adequate muscle relaxation by deep neuromuscular block (NMB) is particularly necessary for MILS to provide a satisfactory surgical field. However, deep NMB for MILS remains controversial, as one-lung ventilation may provide an acceptable surgical field. Then, we will perform a protocol for a systematic review and meta-analysis to identify the efficacy of deep NMB for MILS.

**Methods and analysis** We will search the PubMed, Cochrane Library, Embase, Ovid Medline, Web of Science, Chinese BioMedical Literature, China National Knowledge Infrastructure, VIP and Wanfang databases from inception to March 2022 to identify randomised controlled trials of adult participants undergoing MILS with deep NMB. Studies published in English or Chinese will be considered. The primary outcome will be the surgical conditions according to the surgeon's perspective. Secondary outcomes will be the incidence of perioperative events and perioperative mortality. Heterogeneity will be assessed by the $\chi^2$ test and $I^2$ statistic. Data will be synthesised by both a fixed-effect and a random-effects meta-analysis, with an intention to present the random-effects result if there is no indication of funnel plot asymmetry. Otherwise, metaregression will be used. The Cochrane risk-of-bias tool, trial sequential analysis and Grading of Recommendations Assessment, Development and Evaluation will be used to assess the evidence quality and control the risks of random errors. Funnel plots and Egger's regression test will be used to assess publication bias.

**Ethics and dissemination** Ethical approval was not required for this systematic review protocol. The results will be disseminated through peer-reviewed publications.

**PROSPERO registration number** CRD42021254016.

## STRENGTHS AND LIMITATIONS OF THIS STUDY

⇒ This systematic review protocol according to the Preferred Reporting Items for Systematic Review and Meta-Analysis Protocols guidelines to perform a rigorous risk of bias assessment.

⇒ Trial sequential analysis will be performed to control the risks of false positives by estimating the diversity adjusted information size for the outcomes.

⇒ Funnel plots and Egger's regression test will be applied to assess publication bias.

⇒ Heterogeneity will be assessed by subgroup analysis based on participants' age, body mass index, and type of minimally invasive lung surgery.

## INTRODUCTION

Lung cancer remains the leading cause of cancer death, with an estimated 1.8 million new deaths in 2020, accounting for 18% of the total cancer deaths according to Global Cancer Statistics 2020.[1] Lung cancer is the second most common cancer, with an estimated 2.2 million new cases in 2020, representing 11.4% of all cancer cases.[1] Due to its the high incidence and mortality, the treatment of lung cancer is a global challenge.

Surgical resection remains the primary therapy in the treatment of lung cancer. Since the 1990s, minimally invasive surgical techniques of video-assisted thoracic surgery and robotic-assisted thoracic surgery have been applied in the diagnosis and treatment of intrathoracic diseases.[2–4] Growing experience with minimally invasive lung surgery (MILS), combined with improvements in video technology and instrumentation, has allowed conventional thoracotomy to be gradually replaced by MILS in recent years.[5–8]

Recent literature suggests that MILS was equivalent to open thoracotomy on long-term survival and overall oncologic efficacy, even with a better short-term survival.[9–15] The minimally invasive surgical approach is still the favoured surgical procedure in that it offers many advantages, including less trauma and pain, faster recovery, fewer complications, lower immunological responses and a shorter hospitalisation period.[16–21] In addition, it is

associated with a higher tolerance to postoperative adjuvant therapy and mitigates or ameliorates the postoperative decline in health-related functional status.[22–25]

Adequate muscle relaxation by deep neuromuscular block (NMB) is particularly necessary for minimally invasive surgical techniques.[26–28] MILS involves areas adjacent to major blood vessels and can trigger intraoperative body movement, cough and diaphragm movement.[29] Moreover, the diaphragm is the most resistant muscle to neuromuscular blocking agents (NMBAs), and movement of the diaphragm can interfere with the surgical procedure. Deep NMB can inhibit the response to carinal stimulation and prevent bucking and coughing during surgical procedures.[30–32] In addition, it can reduce the peak pressure and plateau pressure and improve lung compliance and peripheral oxygen saturation during one-lung ventilation.[33]

There is still controversy regarding the clinical benefit of maintaining deep NMB for MILS because deep NMB seems unnecessary, as ribcage provides thoracic support and one-lung ventilation usually provides a satisfactory surgical field. In addition, the risk of residual NMB is estimated to occur in 26%–88% of patients undergoing general anaesthesia, and this incidence is inevitably increased after deep NMB.[34 35] Numerous clinical studies have documented that postoperative residual NMB has the potential risk of increasing the incidence of postoperative pulmonary complications (such as airway obstruction, aspiration and hypoxia), the odds of hospital readmission intensive care unit admission and the hospital length of stay.[36–40]

Hence, the clinical benefits of deep NMB for MILS remain controversial. Therefore, it is necessary to conduct a systematic review and meta-analysis to analyse the clinical efficacy of deep NMB on MILS. The outcomes of this systematic review will provide evidence for better clinical decision-making and possible directions for further clinical trials.

## Objectives

We are performing this protocol of systematic review and meta-analysis to determine the clinical efficacy of deep NMB on the surgical conditions of MILS according to the surgeon's perspective. Patients' postoperative recovery and the incidence of perioperative events will also be identified. Furthermore, trial sequential analysis (TSA) will be applied to confirm the reliability of the results.

## METHODS AND ANALYSIS
### Study design

Our review protocol was registered with PROSPERO 2021 (registration number: CRD 42021254016). This protocol was designed according to the Preferred Reporting Items for Systematic Review and Meta-Analysis Protocols (PRISMA-P) guidelines.[41] The systematic review and meta-analysis will be performed according to the Cochrane Handbook and reported in accordance with the PRISMA

statement.[42 43] The study is anticipated to begin searching in March 2022 and complete in May 2022.

## Inclusion/exclusion criteria for study selection
### Types of studies

We will include all randomised controlled trials (RCTs) involving the efficacy of deep NMB for MILS. Only studies published in English or Chinese will be included.

Studies will be excluded as follows: (1) studies without a control group, compared deep NMB produced by different kinds of NMBAs only; (2) studies with incorrect data obviously, incomplete data or study data that cannot be used for statistical analysis; and (3) studies that were abstracts from conferences, letters, editorials, reviews, observational studies, retrospective studies and duplicate publications.

### Types of participants

Adult participants (≥18 years old) undergoing any kind of MILS (including thoracoscopic surgery, video-assisted thoracic surgery or robotic-assisted thoracic surgery) with deep NMB will be included. No limitations will be defined on participants' characteristics including gender, ethnicity and body mass index (BMI).

### Types of interventions/controls

The intervention group will be the participants who received deep NMB (defined as a train-of-four (TOF) count of zero and a post-tetanic count (PTC)≥1) and intense (profound) NMB (defined as a TOF count=0 and a PTC=0) throughout the MILS.[44]

In the control group, participants had to receive shallow NMB (defined as a TOF count=4 or measured TOF ratio=0.1–0.4), moderate NMB (defined as TOF count=1–3) or without NMBAs throughout the MILS.[44]

### Types of outcome measures

We will perform the meta-analysis only if at least two RCTs have been published in the literature.

## Primary outcomes

The primary outcome will be the surgical conditions of the MILS according to the surgeon's perspective. Surgical conditions were evaluated as a surgical rating scale or the percentage of patients with clinically acceptable surgical conditions (clinically acceptable surgical conditions were defined as Acceptable, Good or Optimal conditions) (table 1).[45]

## Secondary outcomes
### The incidence of perioperative events included the following

► Incidence of intraoperative events: defined as body movement, coughing and breathing against the ventilator (with the aid of airway pressure monitoring and capnography).
► Incidence of postoperative pulmonary complications: defined as the composite of any respiratory infection, respiratory failure, pleural effusion, atelectasis or pneumothorax.

**Table 1** Surgical rating scale (SRS)

| SRS category (scale) | Conditions description |
|---|---|
| Extremely poor conditions (score 1) | The surgeon is unable to work because of coughing or of the inability to obtain a visible field because of inadequate muscle relaxation. |
| Poor conditions (score 2) | There is a visible field, but the surgeon is severely hampered by inadequate muscle relaxation with continuous muscle contractions, movements or both. |
| Acceptable conditions (score 3) | There is a wide visible field but muscle contractions, movements or both occur regularly |
| Good conditions (score 4) | A wide working field with sporadic muscle contractions, movements or both |
| Optimal conditions (score 5) | A wide visible working field without any movement or contractions. |

### Perioperative mortality

► Defined as all-cause death during the operation procedure, within 30 days after surgery, or death during hospitalisation.

### Patients' postoperative recovery

► Recovery time of NMB: defined as the time from administration of the reversal agent to the achievement of a TOF ratio of 0.9.
► Incidence of residual NMB (defined as TOF<0.90 after tracheal extubation/arrival at postanesthesia care unit (PACU).

### Duration of surgery
### Search strategy

We will search English and Chinese electronic databases from inception to March 2022 for published literature.

The English databases included PubMed, Cochrane Library, Embase, Ovid Medline and Web of Science. The Chinese databases included the China National Knowledge Infrastructure, Chinese BioMedical Literature, Wanfang database and VIP Database. We will also scrutinise the reference lists of each study and trial registry database (Clinical Trials.gov and WHO International Clinical Trials Registry Platform) for missing studies and ongoing or unpublished clinical trials. After data extraction, we will ask the corresponding authors of each included literature for more original data to prevent potential missing data as far as possible.

An example of the search strategy used in PubMed is shown in table 2. The search terms will be used as follows: deep neuromuscular block, minimally invasive, thoracoscopic, video assisted, robotic assisted, pulmonary and

**Table 2** Search strategy for PubMed

| No | Search terms |
|---|---|
| #1 | "Neuromuscular blockade"[MeSH] OR neuromusc*[tiab] OR "muscle relaxation" [MeSH] |
| #2 | Deep[tiab] OR profound[tiab] OR intense[tiab] OR extreme[tiab] OR depth[tiab] |
| #3 | "Pulmonary" [Mesh] OR "Lung" [Mesh] OR Pulmonary [tiab] OR Lung [tiab]) |
| #4 | "Surgical Procedures Operative" [Mesh] OR "Microsurgery" [Mesh] OR "Surgical Procedures Minimally Invasive" [Mesh] OR Minimally Invasive Surgery[tiab] OR MIS [tiab] OR Minimal Access Surgical Procedures [tiab]OR Minimal Surgical Procedures[tiab] OR Minimally Invasive Surgical Procedures [tiab] OR Minimal Surgical Procedure[tiab] OR minimally invasive surgical procedure [tiab] OR minimal access surgical procedure[tiab] |
| #5 | "Thoracic surgery, Video-Assisted" [Mesh] or Surgeries, Video-Assisted Thoracic [af] or Surgery, Video-Assisted Thoracic [af] or Thoracic Surgeries, Video-Assisted [af] or Thoracic surgery, Video-Assisted [af] or Video-Assisted Thoracic Surgeries [af] or Video-Assisted Thoracic Surgery [af] or Surgeries, Video-Assisted Thoracoscopic [af] or Surgery, Video-Assisted Thoracoscopic [af] or Thoracoscopic Surgeries, Video-Assisted [af] or Thoracoscopic Surgery, Video-Assisted [af] or Video Assisted Thoracoscopic Surgery [af] or Video Assisted Thoracoscopic Surgeries [af] or Video-Assisted Thoracic Surgery [af] or Video Assisted Thoracic Surgery [af] or Surgery, Thoracic, Video-Assisted [af] or VATS [af] or VATSs [af]. |
| #6 | "Robotics" [MeSH] OR robot* [tiab] OR computer guid*[tiab] OR computer-guid*[tiab] OR computer-assisted[tiab] OR computer assisted [tiab]OR da Vinci [tiab]OR Zeus [tiab]OR telesurgery[tiab] |
| #7 | #1 AND #2 AND #3 |
| #8 | #4 OR #5 OR #6 |
| #9 | "Controlled clinical trial" [Publication Type] OR "randomized controlled trial" [Publication Type] OR "randomized" [Title/Abstract] OR "randomized" [Title/Abstract] OR "Placebo" [Title/Abstract] OR "randomly" [Title/Abstract] OR "Clinical trial" [Title] |
| #10 | "animals" [MeSH] NOT ("human" [MeSH] AND "animals" [MeSH]) |
| #11 | #7and #8 and #9 not #10 |

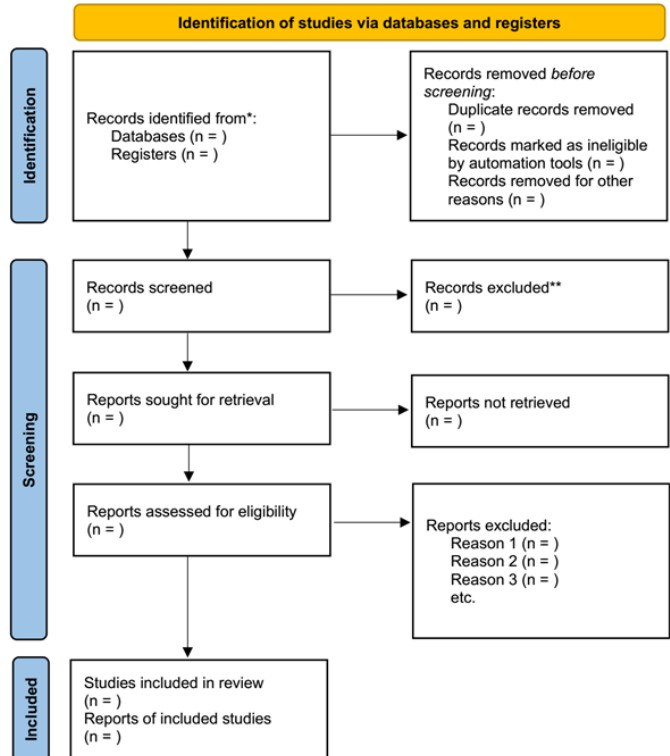

**Figure 1** The Preferred Reporting Items for Systematic Reviews and Meta-Analyses flow diagram.

randomized controlled trial. We will translate the search terms into Chinese for literature research and study identification in Chinese databases. Before the final publication of the systematic review, a latest search in the databases will be performed to check if there are any studies published during the preparation of the systematic review. The preliminary search strategy is listed as online supplemental additional file 1.

### Data collection and analysis
#### Selection of studies
Two reviewers (JZ and JW) will be responsible for screening of the retrieved studies independently. Duplicate studies and those not matching the inclusion criteria will be excluded by reading titles and abstracts briefly. Studies meeting the inclusion criteria will be included after reading the full text of each study thoroughly. Any disagreements will be resolved by consulting a third reviewer (LD) as much as possible. A fourth reviewer (GC) will check out all procedures carefully before confirming the data extraction. The entire study selection process is detailed in the PRISMA flow diagram (figure 1).

### Data extraction
Two reviewers (JZ and LZ) will extract data independently from each included study following a standardised data extraction form (Excel V.2013, Microsoft Inc). Extracted information including participants' demographic data, type of MILS, inclusion and exclusion criteria, level of NMB during MILS (definition and measurement), outcomes (including primary outcomes,

secondary outcomes and exploratory outcomes) and so on. Study design (including randomisation, allocation concealment, blinding, data collection and statistical analysis, outcome reporting) will also be recorded for the subsequent quality assessment. Continuous resulting data will be recorded as the mean±SD, and dichotomous data will be recorded as the proportion of participants with percentages. If necessary, a third reviewer (XD) will cross-check the data to ensure precision. If information and data were missing or incomplete, we will contact authors of the literature to obtain the original data via email. If necessary, numerical data from graphs will be extracted by Adobe Photoshop as described by Gheibi *et al.*[46] A detailed extraction list of information and data is presented in table 3.

### Quality assessment
Two reviewers (JW and LZ) will assess the risk of bias in each included study under the guidance of the Cochrane risk of bias tool independently.[47] We will evaluate the methodology in domains of random sequence generation, allocation concealment, blinding of participants and personnel, blinding of outcome assessment, incomplete outcome data, selective outcome reporting, other risks of bias and overall risk of bias. The risk of bias components will be divided into three levels (low risk, unclear and high risk) according to the checklist item. If all risk of bias domains were scored as having a low risk of bias, the trial was defined as having a low overall risk of bias. If one or more of the bias domains were scored as unclear or high risk of bias, the trial was defined as having a high overall risk of bias. Trials with a low risk of bias in all domains (including sequence generation, allocation concealment, blinding, incomplete data, selective outcome reporting and other risks of bias) will be classified as having a low overall risk of bias. Trials with one or more of these domains scored as unclear or high risk of bias will be defined as having a high overall risk of bias.[48 49] Disagreements, if any, the risk assignment will be settled through arbitration of a third reviewer (GC). Classification of the trials will follow criteria defined in online supplemental additional file 2.

### Measures of treatment effect
Mean differences (MDs) (outcome data reported by same scale) or the standardised MD (outcome data reported by different scales) with 95% CIs will be used for continuous outcome data. While the relative risks (RRs) with 95% CIs will be used for dichotomous data.

### Assessment of heterogeneity
The choice between a fixed-effect and a random-effects meta-analysis based on statistical heterogeneity is not recommended by the Cochrane guidelines.[42] To test the results by the traditional meta-analysis method based on statistical heterogeneity (statistical heterogeneity will be assessed by the standard $\chi^2$ test and $I^2$ test. If p≥0.1 and $I^2$≤50%, the fixed-effects model will be used. If p<0.1 or

**Table 3** Data and information extraction schedule

| Subject | Content |
|---|---|
| Publication information | Name of the first author; contact email; publish year; country; corporate sponsorship. |
| Participant | Sample size; age; sex; height and weight or body mass index; American Society of Anesthesiologists physical status classification levels; type of MILS; inclusion and exclusion criteria if necessary. |
| Intervention | Level of NMB (deep NMB, intense NMB or profound NMB); assessment of the NMB level (equipment of neuromuscular function monitor; monitor position); type of neuromuscular blocking agents (NMBAs); dose and administration of NMBAs; administration of NMBAs antagonist (sugammadex or neostigmine). |
| Control | Level of NMB (moderate NMB; shallow NMB or without NMBAs); assessment of the DNMB (equipment for neuromuscular function monitor; monitor position); type of NMBAs; dose and administration of NMBAs; administration of NMBAs antagonist (sugammadex or neostigmine). |
| Outcome | Primary outcome (surgical rating scale or the percentage of patients with clinically acceptable surgical conditions); secondary outcome measurements (perioperative events; perioperative mortality; patients' postoperative recovery; duration of surgery). |
| Study design | Application of randomisation and blinding; description about allocation concealment; statistical analysis; sample size calculation; outcome reporting. |
| Other information | Intraoperative temperatures; Bispectral Index values; time or condition of tracheal intubation and extubation; type of anaesthesia maintenance technique (inhalation anaesthesia; total intravenous anaesthesia; or both); duration of anaesthesia. |

NMB, neuromuscular block.

$I^2 > 50\%$, the random-effects model will be used), a pragmatic approach will be performed to undertake both a fixed-effect and a random-effects meta-analysis for each outcome, with the intention of presenting the random-effects result if there is no indication of funnel plot asymmetry.[42] If there is an indication of funnel plot asymmetry, then both methods are problematic. It may be reasonable to present both analyses or neither, or to perform a sensitivity analysis in which small studies are excluded or addressed directly using meta-regression. A p<0.05 was assumed to be statistically significant.

**Trial sequential analysis**

We will perform TSA, using the TSA programme V.0.9.5.10 Beta (Copenhagen Trial Unit) to correct the risks of random errors by calculating the required information size (RIS).[50–52] The RIS is defined as the number of participants required in the meta-analysis to detect or reject the intervention effect.[53 54] We will calculate the RIS and information size for each outcome. In addition, the cumulative Z-curve's breach relevant to the TSA monitoring boundaries will be quantified for all outcomes.[53 54]

For continuous outcomes, we will calculate the RIS by the observed SD, an MD of the observed SD/2 (difference of SD/2 is considered clinically meaningful), an alpha (type I error) of 2.5% and a beta (type II error) of 10% for primary and secondary outcomes in the TSA.[55] For dichotomous outcomes, the proportion of participants with an outcome from the control group, a RR reduction/increase of 0.20 (a 20% reduction/increase in RR is considered clinically meaningful) and an alpha (type I error) of 2.5% and a beta (type II error) of 0.10

will be used in the TSA.[56] TSA programme V.0.9.5.10 beta is available at http://www.ctu.dk/tsa.[57]

The diversity adjusted information size (*DIS*) should be calculated, as the RIS might be underestimated. We will use the formula: $DIS = SS/1 - D^2$ ($D^2$: diversity, is the percentage of the variability between trials to the within-trial variance and constitutes the percentage of the variability between trials to the total variance in the meta-analysis. *SS*: sample size in a single randomised clinical trial).[58]

**Subgroup analysis**

We plan to interpret the results through an analysis of subgroups or subsets. If sufficient trials are available (the subgroup analysis will be performed if the variable is reported by at least two RCTs), data from different participants' age, different BMIs and different types of MILSs will be analysed independently.

► Different participants' age (deep NMB for MILS in patients with different ages as follows: 18 years≤patients<65 years; 65 years≤patients<75 years; patients≥75 years).

► Different types of MILS (deep NMB for video-assisted thoracoscopic lung surgery; deep NMB for robotic-assisted thoracoscopic lung surgery).

► Different BMIs (deep NMB for MILS in patients with different BMIs as follows: BMI<25.0 kg/m²; 25.0 kg/m²≤BMI < 30 kg/m²; BMI≥30 kg/m²).

To determine whether a statistically significant subgroup difference was detected, the p value from the test for subgroup differences will be considered. If a significant difference between subgroups is identified

(test for interaction p<0.05), we will report the results for individual subgroups separately.[42]

## Sensitivity analysis

After analysis of subgroups or subsets, sensitivity analysis will be applied to evaluate whether the uncertain assumptions of data and usage could affect the stableness of the combined results. We will exclude low-quality studies (defined as high risk of bias studies according to the Cochrane risk of bias tool assessment), then reanalyse the included studies, as to assess whether there are obvious differences between the combined effects. If necessary, we will remove each included study one by one to detect whether the pooled estimations are stable. Significant changes in the combined results may indicate significant heterogeneity among the included studies.

## Assessment of publication biases

The potential publication bias will be estimated using the funnel plot analysis and Egger's regression test, when more than 10 original studies will be included for an outcome.[59 60] The trim-and-fill analysis will also be applied to confirm any potential publication bias, as it is based on the symmetric pattern of the funnel plot. In the absence of publication bias, the effect sizes of all the studies will be normally distributed around the centre of a funnel plot.[61] Stata/MP V.16.0 (Stata Corp) will be applied to perform the publication biases.

## Grading the quality of evidence

The quality of evidence for all the outcomes will be assessed using the Grading of Recommendations Assessment, Development and Evaluation (GRADE) approach through risk of bias, consistency, objectivity, accuracy and reported bias.[62] The certainty of evidence will be classified as high, moderate, low or very low. According to GRADE, data from RCTs are considered high-quality evidence but can be rated down according to risk of bias, imprecision, inconsistency, indirectness or publication bias.

## Patient and public involvement statement

Patients or the public were not involved in the design, conduct, reporting or dissemination plans of our research.

## DISCUSSION

This systematic review will provide an overview of the current state of evidence on the clinical efficacy of deep NMB for MILS. We will examine the effect of deep NMB on surgical conditions according to the surgeon's perspective. In addition, we will evaluate the efficacy of deep NMB on patients' postoperative recovery and postoperative complications. To our knowledge, this will be the first systematic review on this topic. The results of this systematic review will provide evidence for clinical decision-making on better management of NMB and patient care during MILS.

This systematic review protocol according to the PRISMA-P guidelines. The strengths of our systematic review are as follows: First, we performed a comprehensive search of English and Chinese databases. Second, multivariable analysis (including study quality assessment, subgroup analysis, sensitivity analysis, TSA and Egger's regression test) will be performed to control the confounding bias. Third, two independent reviewers will retrieve literature, extract data, and assess study quality according to the guidelines.

Limitations of our systematic review are as follows: first, studies with different NMBA and NMBA antagonists (sugammadex or neostigmine) will be included, leading to potential heterogeneity. Second, the number of studies with available data for subgroup analyses may be limited. Third, the sample size in each included study may be small. Fourth, studies with high-level evidence such as well-designed RCTs with double-blind designs may be limited. Thus, rigorous meta-analysis methods such as TSA and trim-and-fill analysis will be performed in the data analysis, as to confirm the validity of the outcomes. Finally, it is difficult to define a priori a clinical plausible value of relevant MD and RR increase/decrease during our literature research and clinical experience. Therefore, we defined the clinical plausible value according to the TSA guidelines and the method of sample size calculation.

## ETHICS AND DISSEMINATION

Ethical approval was not required for this systematic review protocol. The findings will be disseminated through peer-reviewed publications.

## Timelines

Formal screening of search results will begin in March 2022. Data extraction will begin in April 2022. The project will be complete in May 2022.

**Contributors** JZ and LD conceived the idea for this systematic review and will conduct the data synthesis. All authors developed the methodology for the systematic review. The manuscript was drafted by JZ and LD, and revised by all authors. XD and GC will screen potential studies, and perform duplicate independent data abstraction. JZ and LZ will undertake risk of bias assessment and assess the evidence quality. All authors contributed to the research and agreed to be responsible for all aspects of the work.

**Funding** The authors have not declared a specific grant for this research from any funding agency in the public, commercial or not-for-profit sectors.

**Competing interests** None declared.

**Patient and public involvement** Patients and/or the public were not involved in the design, or conduct, or reporting, or dissemination plans of this research.

**Patient consent for publication** Not applicable.

**Provenance and peer review** Not commissioned; externally peer reviewed.

**ORCID iD**
Jianqiao Zheng http://orcid.org/0000-0002-8091-1837

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
