## [Reviewer comments · BMJ Open]

ARTICLE DETAILS

TITLE (PROVISIONAL)	Deep neuromuscular block for minimally invasive lung surgery: a protocol for a systematic review with meta-analysis and trial sequential analysis
AUTHORS	Zheng, Jianqiao; Du, Li; Deng, Xiaoqian; Zhang, Lu; Wang, Jia; Chen, Guo

VERSION 1 – REVIEW

REVIEWER	Putzu, Alessandro Geneva University Hospitals
REVIEW RETURNED	15-Nov-2021

GENERAL COMMENTS	I had the pleasure to review a protocol for a systematic review with meta- analysis and trial sequential analysis on deep neuromuscular block for minimally invasive lung surgery. In my opinion, some major revisions are needed to improve clarity and the reproducibility of the study. Major comments: 1- Methods, exclusion criteria. Why will you exclude RCTs comparing DNMB vs NMBAs-free approach? Please report the reason in the discussion.2- Methods, exclusion criteria. “studies with incomplete, incorrect data or the research data could not be used for statistical analysis”. What does it mean? More details are necessary.3- Methods, missing outcome data. How will you manage missing outcome data? I suggest to contact corresponding authors asking for additional data; this may increase the number of eligible trials.4- Methods, outcomes. Please report details on outcomes reporting: e.g. an outcome will be reported if reported by at least 2 RCTs5- Methods, search strategy. Why did you include point #3? This is already included in #4-6.6- Methods, search strategy. I suggest to include other search strategies in the supplementary material.7- Methods, risk of bias assessment. Please include more details on how one study will be rated to be at low, unclear or high risk of bias. Please report more details on “other bias” domain.8- Methods. Why will you include a modified Jaded scale? You will use the Cochrane risk of bias tool. I suggest to remove the Jaded scale.9- Methods. The use of a random or a fixed effects model based on statistical heterogeneity is not recommended by Cochrane guidelines. I suggest to pre-define the use of a random effect model if a high clinical heterogeneity is expected for example (or heterogeneity in assessment scales). “A pragmatic approach is to plan to undertake both a fixed-effect and a random-effects meta-analysis, with an intention to present the random-effects result if
--

	there is no indication of funnel plot asymmetry” (Cochrane handbook 10.10.4.1 Fixed or random effects?#section-10-10-4-1) 10- Methods, trial sequential analysis. It is not recommended to use mean difference of the observed dataset. You should define a priori a clinical plausible and relevant mean difference and relative risk increase/decrease to use in the TSA. 11- Methods, trial sequential analysis. The TSA will be diversity adjusted? 12- Methods, “For continuous outcome data, the mean differences (MDs) or the standardised mean difference (SMDs) with 95% confidence intervals (CIs)”. Please define when you will use the MD or the SMD. 13- Methods, subgroup analyses. Please report more details on subgroup analyses reporting: e.g. the subgroup analysis will be performed if the variable is reported by at least 2 RCTs. 14- Methods, subgroup analyses. How will you assess interaction between group? Please report the significance level of the p value for interaction. 15- Methods, sensitivity analysis. Please define low quality study (e.g.; RCTs with high and unclear risk of bias?). 16- Methods. An important point that should be included is a GRADE certainty of evidence assessment. 17- Methods. I suggest to include a PRISMA 2020 checklist and PRISMA 2020 flow-chart. 18- Methods. Please define the significance level for p value. Minor comments: 1- Abstract. Please report primary outcome of the study. 2- Abstract. I suggest to include more details on inclusion criteria. 3- Abstract. No need to cite the software in the abstract. 4- Methods, outcomes. I suggest to include mortality as secondary outcome or as exploratory outcome.
--	--

REVIEWER	Brull, Sorin Mayo Clinic, Anesthesiology and Perioperative Medicine Dr. Brull has intellectual property assigned to Mayo Clinic (Rochester, MN); has received research support (funds to Mayo Clinic) from Merck & Co., Inc. (Kenilworth, NJ) and is a consultant for Merck & Co., Inc.; is a principal, shareholder and Chief Medical Officer in Senzime AB (publ) (Uppsala, Sweden); and a member of the Scientific/Clinical Advisory Boards for The Doctors Company (Napa, CA); Coala Life Inc. (Irvine, CA); NMD Pharma (Aarhus, Denmark); and Takeda Pharmaceuticals (Cambridge, MA)
REVIEW RETURNED	23-Dec-2021

GENERAL COMMENTS	This is a very interesting, clinically relevant, and topical proposed protocol for performing a systematic review (meta-analysis and trial sequential analysis) of the utility of deep neuromuscular block in minimally invasive thoracic surgery. The authors are to be congratulated for undertaking this topic, since it is important but under-reported; in fact, this reviewer only found less than 10 articles in which deep neuromuscular block had been used in lung/thoracic surgery. Have the authors performed a brief literature review of the topic to ensure such studies are available? Despite the novelty and importance of the topic, the reviewer has reservations about the planned meta-analysis. The title, “Deep
--

	neuromuscular block for minimally invasive lung surgery” suggests that the effectiveness of deep block will be assessed. The methods, however, describe deep neuromuscular block compared to shallow and moderate block (using an unconventional definition of shallow and moderate block). The reviewer recommends that the authors refer to a set of recognized definitions of the depth of neuromuscular block (see, for instance, PMID: 27820709). The other issue is the authors’ use of “on demand NMB,” defined as antagonists(!) given on-demand; and “standard NMB,” defined as antagonists “given as indicated.” This reviewer does not follow the rationale for including these unconventional definitions. The primary outcome (surgical conditions “according to the surgeon’s perspective”) appears somewhat subjective (despite the use of a rating scale) and dependent on factors (surgeon training, expertise, experience, etc.) not associated with the anesthetic depth of neuromuscular block. Additionally, the secondary aims do not quite make sense to the reviewer: while I could understand, perhaps, perioperative arterial blood gas analyses, the rest of the outcome parameters appear random and unlikely to result in usable/comparable data (for instance, postoperative pain is multifactorial). Specific Comments Page=P; Line Number=L P6L14 – Suggest, “became” P6L22 – Suggest, “field” P6L22 – Please provide support for the contention that one-lung ventilation does not require deep block. P6 – Here and throughout, please limit the number of non-standard abbreviation (such as DNMB; MILS; MIS; RATS; VATS, etc.) – such abbreviations make the manuscript very difficult to read. P2L43 – Why have a cut-off of September 2021, and not include literature from the past few months? After all, this is a proposed meta-analysis. P4L12 – Suggest, “accounting for 18%...” P4L17 – Suggest, “common”; please rephrase the last sentence (L20-22) P4 – MIS and MILS are used interchangeably – what is the difference between the two? P5L14 – Please provide literature support for the (wrong) assertion that deep neuromuscular block is “mandatory for most surgical procedures.” P7L7-10 – Will the search require 4 months? P7L28-30 – Why exclude studies that investigated deep block
--	--

	achieved “by different kinds of NMBAAs”? The authors are not investigating pharmacokinetics/dynamics, so would deep neuromuscular block produced by rocuronium, for instance, be different from that produced by vecuronium atracurium, cisatracurium, etc.? P7L30 – Similarly, why exclude studies that may have compared deep block with “NMBA-free” surgery? What is the rationale? P7L59 – The statement, “In the intervention group” suggests that there may be a “non-intervention” group (a group in which patients did NOT receive neuromuscular blocking agents). According to the exclusion criteria, though, these studies would be excluded from analysis. Please explain. P8L30-41 – Is the surgical rating scale validated? P9L27-33 – Why is the recovery time a secondary aim? P19L25-28 – Why is the search in English and Chinese databases a strength of the systematic review? P19L43-46 – Why would the inclusion of different antagonists (sugammadex and neostigmine) be considered a “limitation” of this meta-analysis of the utility of deep block? Also, “neostigmine and sugammadex” are generic names, and they should not be capitalized. P20 – Timelines – It appears that the screening of search results has already started – can the authors confirm?
--	--

VERSION 1 – AUTHOR RESPONSE

Reviewer: 1

Dear Dr. Alessandro Putzu, Geneva University Hospitals

Thank you very much for your reviewing.

Major comments:

1- **Methods, exclusion criteria.** Why will you exclude RCTs comparing DNMB vs NMBAAs-free approach? Please report the reason in the discussion.

Revised. “NMBA-free” as the control group should not excluded.

2- **Methods, exclusion criteria.** “studies with incomplete, incorrect data or the research data could not be used for statistical analysis”. What does it mean? More details are necessary.

Incomplete data: means the eligible data was reported in other outcome forms which cannot be used in the data synthesis of the meta-analysis, and the original eligible data cannot be obtained after contacting the corresponding authors. So, when we found the incorrect data in the literature search, we will exclude these literatures.

Incorrect data: in our previous experience of meta-analysis, when we extract the data, we found some data was labeled with wrong units (such as milligrams incorrectly labeled with micrograms) or inconsistent unit (such as the unit of $P_{ET}CO_2$ was cmH_2O , but the description was mmHg in the literature), inconsistent participant number in the same group after the total number minus the participant of drop out and withdrawal, and obvious incorrect writing (such as the NBP was

980mmHg, maybe the correct data is 98 mmHg). Most of the incorrect data cases were founded in the literature from Chinese literature databases. Prior to final publication of our meta-analysis, we will perform a new search in the databases to check if any studies were published during the elaboration of the systematic review. So, when we found the incorrect data in the literature search, we will exclude these literatures.

The research data could not be used for statistical analysis: the research data about the outcome was not the primary outcome or the secondary outcome of our meta-analysis.

3- Methods, missing outcome data. How will you manage missing outcome data? I suggest to contact corresponding authors asking for additional data; this may increase the number of eligible trials.

We agree with your opinions. After data extraction, we will ask the corresponding authors of the included literature for more grey literature to avoid potential missing data as much as possible.

4- Methods, outcomes. Please report details on outcomes reporting; e.g. an outcome will be reported if reported by at least 2 RCTs

Revised

5- Methods, search strategy. Why did you include point #3? This is already included in #4-6.

#3 related to the lung or pulmonary;

#4-6 related to the minimally invasive surgical procedure, such as video assisted thoracic surgery, computer-assisted OR robot-assisted surgery, it will including pulmonary surgery and e esophageal surgery. So, we used #3 and #4-6 to searching the minimally invasive lung surgery.

6- Methods, search strategy. I suggest to include other search strategies in the supplementary material.

Revised

7- Methods, risk of bias assessment. Please include more details on how one study will be rated to be at low, unclear or high risk of bias. Please report more details on "other bias" domain.

Revised. We agree with your opinion and use the Cochrane risk of bias tool to estimate the risk of bias assessment. We will provide the Assessment of risk of bias as online supplemental additional file. Assessment of risk of bias in respect of: Random sequence generation; Allocation concealment; Blinding of participants and treatment providers; Blinding of outcome assessment; Incomplete outcome data; Selective outcome reporting; Overall risk of bias. The risk of bias components will be scored as three levels (low risk, unclear and high risk) in accordance with the item in the checklist.

Random sequence generation

- *Low risk:* If sequence generation was achieved using computer random number generator or a random number table. Drawing lots, tossing a coin, shuffling cards, and throwing dice were also considered adequate if performed by an independent adjudicator.
- *Unclear risk:* If the method of randomisation was not specified, but the trial was still presented as being randomised.
- *High risk:* If the allocation sequence is not randomised or only quasi-randomised. These trials will be excluded.

Allocation concealment

- *Low risk:* If the allocation of patients was performed by a central independent unit, onsite locked computer or identical-looking numbered sealed envelopes.
- *Uncertain risk:* If the trial was classified as randomised but the allocation concealment process was not described.
- *High risk:* If the allocation sequence was familiar to the investigators who assigned participants.

Blinding of participants and treatment providers

- *Low risk:* If the participants and the treatment providers were blinded to intervention allocation and this was described.

- *Uncertain risk*: If the procedure of blinding was insufficiently described.
- *High risk*: If blinding of participants and the treatment providers was not performed.

Blinding of outcome assessment

- *Low risk of bias*: If it was mentioned that outcome assessors were blinded and this was described.
- *Uncertain risk of bias*: If it was not mentioned if the outcome assessors in the trial were blinded or the extent of blinding was insufficiently described.
- *High risk of bias*: If no blinding or incomplete blinding of outcome assessors was performed.

Incomplete outcome data

- *Low risk of bias*: If missing data were unlikely to make treatment effects depart from plausible values. This could be either (1) there were no drop-outs or withdrawals for all outcomes, or (2) the numbers and reasons for the withdrawals and drop-outs for all outcomes were clearly stated and could be described as being similar to both groups. Generally, the trial is judged as at a low risk of bias due to incomplete outcome data if drop-outs are less than 5%. However, the 5% cut-off is not definitive.
- *Uncertain risk of bias*: If there was insufficient information to assess whether missing data were likely to induce bias on the results.
- *High risk of bias*: If the results were likely to be biased due to missing data either because the pattern of drop-outs could be described as being different in the two intervention groups or the trial used improper methods in dealing with the missing data (e.g. last observation carried forward).

Selective outcome reporting

- *Low risk of bias*: If a protocol was published before or at the time the trial was begun and the outcomes specified in the protocol were reported on. If there is no protocol or the protocol was published after the trial has begun, reporting of serious adverse events will grant the trial a grade of low risk of bias.
- *Uncertain risk of bias*: If no protocol was published and the outcome of serious adverse events were not reported on.
- *High risk of bias*: If the outcomes in the protocol were not reported on.

Other risks of bias

- *Low risk of bias*: If the trial appears to be free of other components (for example, academic bias or for-profit bias) that could put it at risk of bias.
- *Unclear risk of bias*: If the trial may or may not be free of other components that could put it at risk of bias.
- *High risk of bias*: If there are other factors in the trial that could put it at risk of bias (for example, authors conducted trials on the same topic, for-profit bias, etc.).

Overall risk of bias

- *Low risk of bias*: The trial will be classified as overall 'low risk of bias' only if all of the bias domains described in the above paragraphs are classified as 'low risk of bias'.
- *High risk of bias*: The trial will be classified as 'high risk of bias' if any of the bias risk domains described in the above are classified as 'unclear' or 'high risk of bias'.
- We will assess the domains 'blinding of outcome assessment', 'incomplete outcome data', and 'selective outcome reporting' for each outcome result. Thus, we can assess the bias risk for each outcome assessed in addition to each trial. Our primary conclusions will be based on the results of our primary outcome results with overall low risk of bias. Both our primary and secondary conclusions will be presented in the summary of findings tables.

Other bias will be other factors in the trial that could put it at risk of bias, such as academic bias or for-profit bias, or authors conducted different trials on the same or similar topic, or authors published articles on the same topic with different languages, such as the English and Chinese, or authors published articles on the same topic with different sample, or trials without registration.

8- Methods. Why will you include a modified Jaded scale? You will use the Cochrane risk of bias tool. I suggest to remove the Jaded scale.

We agree with your opinion. We removed the Jaded scale and revised to use the Cochrane risk of bias tool to estimate the risk of bias assessment.

9- Methods. The use of a random or a fixed effects model based on statistical heterogeneity is not recommended by Cochrane guidelines. I suggest to pre-define the use of a random effect model if a high clinical heterogeneity is expected for example (or heterogeneity in assessment scales). "A pragmatic approach is to plan to undertake both a fixed-effect and a random-effects meta-analysis, with an intention to present the random-effects result if there is no indication of funnel plot asymmetry" (Cochrane handbook 10.10.4.1 Fixed or random effects? # section-10-10-4-1)

Revised. Thank you very much for your recommendation, we will perform a pragmatic approach and the traditional meta-analysis method based on statistical heterogeneity to justify the results.¹

[1] Deeks JJ, Higgins JPT, Altman DG (editors). Chapter 10: Analysing data and undertaking meta-analyses. In: Higgins JPT, Thomas J, Chandler J, Cumpston M, Li T, Page MJ, Welch VA (editors). Cochrane Handbook for Systematic Reviews of Interventions version 6.2 (updated February 2021). Cochrane, 2021. Available from www.training.cochrane.org/handbook.

10- Methods, trial sequential analysis. It is not recommended to use mean difference of the observed dataset. You should define a priori a clinical plausible and relevant mean difference and relative risk increase/decrease to use in the TSA.

Thank you very much for your recommendation, and we cannot agree more. But during our literature research and clinical experience, it is hard to define a priori a clinical plausible value of relevant mean difference and relative risk increase/decrease. So, we define the clinical plausible value according to the TSA guideline and the method of sample size calculation.

11- Methods, trial sequential analysis. The TSA will be diversity adjusted?

Thank you very much for your recommendation. A really helpful advice. Diversity adjusted information size (DIS) should be calculated, as the required information size might be underestimated. We will use the formula: $DIS = SS / (1 - D^2)$ (D^2 : Diversity, is the percentage of the variability between trials to the within-trial variance and constitutes the percentage of the variability between trials to the total variance in the meta-analysis. SS: Sample size in a single randomized clinical trial).¹

[1] Wetterslev J, Thorlund K, Brok J, Gluud C. Estimating required information size by quantifying diversity in random-effects model meta-analyses. *BMC Med Res Methodol.* 2009; 9:86

12- Methods, "For continuous outcome data, the mean differences (MDs) or the standardised mean difference (SMDs) with 95% confidence intervals (CIs)". Please define when you will use the MD or the SMD.

Revised. The mean differences (MDs) will be used when continuous outcome data reported by same scale, the standardised mean difference (SMDs) will be used when continuous outcome data reported by different scales.

13- Methods, subgroup analyses. Please report more details on subgroup analyses reporting: e.g. the subgroup analysis will be performed if the variable is reported by at least 2 RCTs.

Revised.

14- Methods, subgroup analyses. How will you assess interaction between group? Please report the significance level of the p value for interaction.

According to Donegan et al.,¹ when interpreting subgroup analyses, review authors could consider five key criteria: 1) whether a statistically significant subgroup difference (interaction) was detected; 2) the covariate distribution (i.e. the number of trials and participants contributing to each subgroup); 3) the plausibility of the interaction or lack of interaction; 4) the importance of the interaction or lack of interaction; 5) the possibility of confounding. To determine whether a statistically significant subgroup difference was detected, the p value from the test for subgroup differences ought to be considered. If a significant difference between subgroups is identified (test for interaction $p < 0.05$), we will report the results for individual subgroups separately.

[1] Donegan S, Williams L, Dias S, Tudur-Smith C, Welton N. Exploring treatment by covariate interactions using subgroup analysis and meta-regression in cochrane reviews: a review of recent practice. PLoS One. 2015;10(6): e0128804

15- Methods, sensitivity analysis. Please define low quality study (e.g.; RCTs with high and unclear risk of bias?).

Revised. Study quality will be defined as the low quality (high risk of bias), high quality (low risk of bias) according to the Cochrane risk of bias tool assessment. Low risk of bias: The trial will be classified as overall 'low risk of bias' only if all of the bias domains described in the above paragraphs are classified as 'low risk of bias'. High risk of bias: The trial will be classified as 'high risk of bias' if any of the bias risk domains described in the above are classified as 'unclear' or 'high risk of bias'.

16- Methods. An important point that should be included is a GRADE certainty of evidence assessment.

Revised. Grading the quality of evidence was added in the methods part. The quality of evidence for all the outcomes will be assessed using the GRADE approach through risk of bias, consistency, objectivity, accuracy and reported bias. The certainty of evidence will be classified as high, moderate, low or very low. According to GRADE, data from randomized controlled trials are considered high quality evidence but can be rated down according to risk of bias, imprecision, inconsistency, indirectness or publication bias.

17- Methods. I suggest to include a PRISMA 2020 checklist and PRISMA 2020 flow-chart.

Revised, we changed to the PRISMA 2020 flow-chart, but we still used the PRISMA -P-checklist as our manuscript is a protocol

18- Methods. Please define the significance level for p value.

Revised

Minor comments:

1- Abstract. Please report primary outcome of the study.

Revised. We performed a brief literature review of the topic and only found four studies. This circumstance may reflect the infancy of research in the deep neuromuscular block for minimally invasive lung surgery. In the era of VATS and ERAS, it is necessary to explore the best clinical decision making on the management of intraoperative NMB. We believe that there will be more RCTs on deep neuromuscular block for minimally invasive lung surgery, which can be used for meta-analysis. In addition, we will plan to design a RCT on this topic in future, which could also to provide data analysis and evidence for this meta-analysis. So, we want to publish this protocol first as to avoid duplication of this topic by other researchers.

2- Abstract. I suggest to include more details on inclusion criteria.

Revised

3- Abstract. No need to cite the software in the abstract.

Revised

4- Methods, outcomes. I suggest to include mortality as secondary outcome or as exploratory outcome.

Revised.

Reviewer: 2

Dear Dr. Sorin Brull, Mayo Clinic

Thank you very much for your reviewing.

Comments to the Author:

The authors are to be congratulated for undertaking this topic, since it is important but under-reported; in fact, this reviewer only found less than 10 articles in which deep neuromuscular block had been used in lung/thoracic surgery. Have the authors performed a brief literature review of the topic to ensure such studies are available?

We performed a brief literature review of the topic and only found four studies. This circumstance may reflect the infancy of research in the deep neuromuscular block for minimally invasive lung surgery. In the era of VATS and ERAS, it is necessary to explore the best clinical decision making on the management of intraoperative NMB. We believe that there will be more RCTs on deep neuromuscular block for minimally invasive lung surgery, which can be used for meta-analysis. In addition, we will plan to design a RCT on this topic in future, which could also to provide data analysis and evidence for this meta-analysis. So, we want to publish this protocol first as to avoid duplication of this topic by other researchers.

Despite the novelty and importance of the topic, the reviewer has reservations about the planned meta-analysis. The title, "Deep neuromuscular block for minimally invasive lung surgery" suggests that the effectiveness of deep block will be assessed. The methods, however, describe deep neuromuscular block compared to shallow and moderate block (using an unconventional definition of shallow and moderate block). The reviewer recommends that the authors refer to a set of recognized definitions of the depth of neuromuscular block (see, for instance, PMID: 27820709).

Revised, the depth of neuromuscular block was revised to the definitions from this literature.

The other issue is the authors' use of "on demand NMB," defined as antagonists(!) given on-demand; and "standard NMB," defined as antagonists "given as indicated." This reviewer does not follow the rationale for including these unconventional definitions.

Revised and deleted these unconventional definitions of NMB from control group.

The primary outcome (surgical conditions "according to the surgeon's perspective") appears somewhat subjective (despite the use of a rating scale) and dependent on factors (surgeon training, expertise, experience, etc.) not associated with the anesthetic depth of neuromuscular block. Additionally, the secondary aims do not quite make sense to the reviewer: while I could understand, perhaps, perioperative arterial blood gas analyses, the rest of the outcome parameters appear random and unlikely to result in usable/comparable data (for instance, postoperative pain is multifactorial).

Revised. The surgical rating scale is really somewhat subjective and dependent on the characteristics of the surgeon. But, during our literature searching about "deep NMB for Surgery", in most clinical studies and meta-analysis, the surgical rating scale was still defined as the primary outcome¹⁻⁵.

- [1] Raval AD, Deshpande S, Rabar S, Koufopoulou M, Neupane B, Iheanacho I, Bash LD, Horrow J, Fuchs-Buder T. Does deep neuromuscular blockade during laparoscopy procedures change patient, surgical, and healthcare resource outcomes? A systematic review and meta-analysis of randomized controlled trials. *PLoS One*. 2020 Apr 16;15(4):e0231452.
- [2] Aceto P, Perilli V, Modesti C, Sacco T, De Cicco R, Ceaichisciuc I, Sollazzi L. Effects of deep neuromuscular block on surgical workspace conditions in laparoscopic bariatric surgery: a systematic review and meta-analysis of randomized controlled trials. *Minerva Anesthesiol*. 2020 Sep;86(9):957-964.
- [3] Wei Y, Li J, Sun F, Zhang D, Li M, Zuo Y. Low intra-abdominal pressure and deep neuromuscular blockade laparoscopic surgery and surgical space conditions: A meta-analysis. *Medicine (Baltimore)*. 2020 Feb;99(9):e19323.
- [4] Park SK, Son YG, Yoo S, Lim T, Kim WH, Kim JT. Deep vs. moderate neuromuscular blockade during laparoscopic surgery: A systematic review and meta-analysis. *Eur J Anaesthesiol*. 2018 Nov;35(11):867-875.
- [5] Bruintjes MH, van Helden EV, Braat AE, Dahan A, Scheffer GJ, van Laarhoven CJ, Warlé MC. Deep neuromuscular block to optimize surgical space conditions during laparoscopic surgery: a systematic review and meta-analysis. *Br J Anaesth*. 2017 Jun 1;118(6):834-842.

We agree with your opinion and the secondary outcomes were changed to:

(1) Duration of surgery;

(2) The incidence of perioperative events included the following:

► Incidence of intraoperative events: defined as body movement, coughing, and breathing against the ventilator (with the aid of airway pressure monitoring and capnography).

► Incidence of postoperative pulmonary complications: defined as the composite of any respiratory infection, respiratory failure, pleural effusion, atelectasis, or pneumothorax.

(3) Perioperative mortality: ► defined as all-cause death during operation procedure, within 30 days after surgery, or death during hospitalization.

(4) Patients' postoperative recovery: ► Recovery time of NMB: defined as the time from administration of the reversal agent to the achievement of a TOF ratio of 0.9.

Specific Comments

P6L14 – Suggest, “became”

Revised

P6L22 – Suggest, “field”

Revised

P6L22 – Please provide support for the contention that one-lung ventilation does not require deep block.

During our literature searching, we found some clinical studies have confirmed that target-controlled infusion

of remifentanil without muscle relaxants allows acceptable surgical conditions during thoracotomy performed under sevoflurane anesthesia.¹ To avoid the negative side effects of tracheal intubation and general anesthesia, non-intubated VATS techniques began to be considered as an alternative to general anesthesia with intubation,² but it remains controversy.³

[1] El-Tahan MR, Regal M. Target-Controlled Infusion of Remifentanil Without Muscle Relaxants Allows Acceptable Surgical Conditions During Thoracotomy Performed Under Sevoflurane Anesthesia. *J Cardiothorac Vasc Anesth.* 2015 Dec;29(6):1557-66.

[2] Guo Z, Shao W, Yin W, Chen H, Zhang X, Dong Q, Liang L, Wang W, Peng G, He J. Analysis of feasibility and safety of complete video-assisted thoracoscopic resection of anatomic pulmonary segments under non-intubated anesthesia. *J Thorac Dis.* 2014 Jan;6(1):37-44.

[3] Janík M, Juhos P, Lučenič M, Tarabová K. Non-intubated Thoracoscopic Surgery-Pros and Cons. *Front Surg.* 2021 Dec 6;8:801718.

P6 – Here and throughout, please limit the number of non-standard abbreviation (such as DNMB; MILS; MIS; RATS; VATS, etc.) – such abbreviations make the manuscript very difficult to read.

Revised.

P2L43 – Why have a cut-off of September 2021, and not include literature from the past few months? After all, this is a proposed meta-analysis.

Revised, sorry for the wrong description. “The study is expected to begin searching in September 2021 and end in December 2021”, we submitted our protocol in 26-Aug-2021 and want to start the literature searching in September 2021, then finish this meta-analysis in December 2021. This was our original plan. However, we want to complete this meta-analysis under the guidance of the reviewer of the journal, so we have been waiting for the reviewer's opinion after submission.

P4L12 – Suggest, “accounting for 18%...”

Revised.

P4L17 – Suggest, “common”; please rephrase the last sentence (L20-22)

Revised.

P4 – MIS and MILS are used interchangeably – what is the difference between the two?

Revised.

P5L14 – Please provide literature support for the (wrong) assertion that deep neuromuscular block is “mandatory for most surgical procedures.”

Revised, sorry for the wrong description.

P7L7-10 – Will the search require 4 months?

Revised, sorry for the wrong description. “The study is expected to begin searching in September 2021 and end in December 2021”, we submitted our protocol in 26-Aug-2021 and want to start the

literature searching in September 2021. After finish this meta-analysis in December 2021, we want to complete this meta-analysis under the guidance of the reviewer of the journal, so we have been waiting for the reviewer's opinion after submission. In fact, our meta-analysis has not completed and only the literature searching was completed. Next procedure, we will complete this meta-analysis according to the reviewer's comment. Therefore, we will revise the anticipated completion date in the timelines of our manuscript, simultaneously, we will update the anticipated completion date on the website of PROSPERO.

P7L28-30 – Why exclude studies that investigated deep block achieved “by different kinds of NMBAs”? The authors are not investigating pharmacokinetics/dynamics, so would deep neuromuscular block produced by rocuronium, for instance, be different from that produced by vecuronium atracurium, cisatracurium, etc.?

Revised, sorry for the wrong description. When deep NMB was used in both control group and intervention group, but the deep neuromuscular block produced by different kinds of NMBAs, such as the studied just compare the efficiency of deep NMB by different kinds of NMBAs. These studies will be excluded.

P7L30 – Similarly, why exclude studies that may have compared deep block with “NMBA-free” surgery? What is the rationale?

Revised. “NMBA-free” as the control group should not excluded, it is really irrational.

P7L59 – The statement, “In the intervention group” suggests that there may be a “non-intervention” group (a group in which patients did NOT receive neuromuscular blocking agents). According to the exclusion criteria, though, these studies would be excluded from analysis. Please explain.

Revised. We apologize for the confusion caused by our inaccurate description.

In the previous version, the inclusion criteria will be: Adult participants (age ≥ 18 years old) undergoing any kind of MILS (including thoracoscopic surgery, video-assisted thoracic surgery or robotic-assisted thoracic surgery) with deep NMB will be included.

The exclusion criteria will be: (1) studies without a control group; (2) studies that only compared deep NMB produced by different kinds of NMBAs or compared deep NMB surgery with NMBAs-free surgery; (3) studies with incomplete, incorrect data or research data that could not be used for statistical analysis; and (4) studies that were abstracts from conferences, editorials, duplicate publications, letters, reviews, observational studies, and retrospective studies.

We agree with your opinion “NMBA-free” as the control group should not excluded, and we changed our exclusion criteria: (1) studies without control group, compared deep NMB produced by different kinds of NMBAs only; (2) studies with incomplete, incorrect data or research data that could not be used for statistical analysis; and (3) studies that were abstracts from conferences, editorials, duplicate publications, letters, reviews, observational studies, and retrospective studies.

P8L30-41 – Is the surgical rating scale validated?

The surgical rating scale is really somewhat subjective and dependent on the characteristics of the surgeon. But, during our literature searching about “deep NMB for Surgery”, in most clinical studies and meta-analysis, the surgical rating scale was still defined as the primary outcome¹⁻⁵. So, we think the surgical rating scale still remains the preferred primary outcome in our meta-analysis.

[1] Raval AD, Deshpande S, Rabar S, Koufopoulou M, Neupane B, Iheanacho I, Bash LD, Horrow J, Fuchs-Buder T. Does deep neuromuscular blockade during laparoscopy procedures change patient, surgical, and healthcare resource outcomes? A systematic review and meta-analysis of randomized controlled trials. *PLoS One*. 2020 Apr 16;15(4):e0231452.

[2] Aceto P, Perilli V, Modesti C, Sacco T, De Cicco R, Ceaichisciuc I, Sollazzi L. Effects of deep neuromuscular block on surgical workspace conditions in laparoscopic bariatric surgery: a

systematic review and meta-analysis of randomized controlled trials. *Minerva Anesthesiol.* 2020 Sep;86(9):957-64.

- [3] Wei Y, Li J, Sun F, Zhang D, Li M, Zuo Y. Low intra-abdominal pressure and deep neuromuscular blockade laparoscopic surgery and surgical space conditions: A meta-analysis. *Medicine (Baltimore).* 2020 Feb;99(9):e19323.
- [4] Park SK, Son YG, Yoo S, Lim T, Kim WH, Kim JT. Deep vs. moderate neuromuscular blockade during laparoscopic surgery: A systematic review and meta-analysis. *Eur J Anaesthesiol.* 2018 Nov;35(11):867-875.
- [5] Brintjes MH, van Helden EV, Braat AE, Dahan A, Scheffer GJ, van Laarhoven CJ, Warlé MC. Deep neuromuscular block to optimize surgical space conditions during laparoscopic surgery: a systematic review and meta-analysis. *Br J Anaesth.* 2017 Jun 1;118(6):834-842.

P9L27-33 – Why is the recovery time a secondary aim?

Deep NMB also has some drawbacks including the potential for postoperative residual curarization. Although antagonists (sugammadex and neostigmine) decreased the incidence of residual NMB significantly, residual NMB still exists. Residual muscle relaxation significantly extends the time required for extubation and postoperative monitoring, may cause airway obstruction, respiratory depression and hypoxemia. In this situation, residual muscle relaxation will increase the recovery time and medical expenses. So, the recovery time was defined as one of the secondary outcomes.

P19L25-28 – Why is the search in English and Chinese databases a strength of the systematic review?

Chinese databases are not considered during the literature searching in many systematic reviews. Therefore, some Chinese articles were excluded. Although the overall quality of Chinese literatures is not high, there are still some high-quality literatures that can be used for data analysis. So, searching Chinese database is beneficial to enlarge the sample size of meta-analysis and increase the quality of evidence.

P19L43-46 – Why would the inclusion of different antagonists (sugammadex and neostigmine) be considered a “limitation” of this meta-analysis of the utility of deep block? Also, “neostigmine and sugammadex” are generic names, and they should not be capitalized.

In our literature searching, a cochrane database systematic review compared sugammadex and neostigmine in reversing deep NMB from post-tetanic count (PTC) 1 to 5 to TOFR > 0.9, sugammadex 4 mg/kg was 2.9 minutes, neostigmine 0.07 mg/kg was 48.8 minutes. Sugammadex was 45.78 minutes faster (52.15 to 39.41 minutes faster) than neostigmine¹. The heterogeneity could be high when the reversing time of sugammadex and neostigmine are counted together. In addition, subgroup analysis might not be possible as the limited number of included articles. In this situation, heterogeneity will influence the result and the evidence quality. It might be a “limitation” of this meta-analysis.

- [1] Hristovska AM, Duch P, Allingstrup M, Afshari A. Efficacy and safety of sugammadex versus neostigmine in reversing neuromuscular blockade in adults. *Cochrane Database Syst Rev.* 2017;8(8):CD012763. doi: 10.1002/14651858.CD012763.

P20 – Timelines – It appears that the screening of search results has already started – can the authors confirm?

Revised, we submitted our protocol in 26-Aug-2021, then we want to start the literature searching in September 2021, and finish this meta-analysis in December 2021. But, we want to complete this meta-analysis under the guidance of the reviewer of the journal, so we have been waiting for the reviewer's opinion after submission. In fact, the screening of search results has not already started and only the literature searching was completed. Next procedure, we will complete this meta-analysis according to the reviewer's comment. Therefore, we will revise the anticipated completion date in the timelines of our manuscript, simultaneously, we will update the anticipated completion date on the website of PROSPERO.

VERSION 2 – REVIEW

REVIEWER	Putzu, Alessandro Geneva University Hospitals
REVIEW RETURNED	23-Feb-2022

GENERAL COMMENTS	Thank you for your great work on the manuscript. I have some minor comments: 1- Abstract. "Data will be synthesized by either fixed-effects or random-effects models according to the I2 value." This is correct? 2- Trial sequential analysis. I suggest to define a priori a clinical plausible and relevant mean difference to use in the TSA. If it is not the case, it should be reported as a study limitation. 3- Risk of bias. I suggest to report in details the items included in the 'other bias' domain (e.g.; financial and non-financial conflict of interest), avoiding general statements such as 'etcetera'. 4- Manuscript. I suggest to further improve the clarity and the quality of the English throughout your manuscript.
--

REVIEWER	Brull, Sorin Mayo Clinic, Anesthesiology and Perioperative Medicine Dr. Brull has intellectual property assigned to Mayo Clinic (Rochester, MN); has received research support (funds to Mayo Clinic) from Merck & Co., Inc. (Kenilworth, NJ) and is a consultant for Merck & Co., Inc.; is a principal, shareholder and Chief Medical Officer in Senzime AB (publ) (Uppsala, Sweden); and a member of the Scientific/Clinical Advisory Boards for The Doctors Company (Napa, CA); Coala Life Inc. (Irvine, CA); NMD Pharma (Aarhus, Denmark); and Takeda Pharmaceuticals (Cambridge, MA)
REVIEW RETURNED	24-Feb-2022

GENERAL COMMENTS	Please review and correct the depth of block definitions listed in the manuscript pages 7-8: deep NMB is defined as TOF count of zero AND (not "or") a PTC ≥ 1; intense NMB is defined as TOF count=0 AND (not "or") PTC=0; shallow NMB is defined as TOF count=4 AND (not "or") TOF ratio=0.1-0.4. Page 8: Types of Outcome Measures. Suggest: "We will perform the meta-analysis only if at least...xxx number of studies have been published in the literature." - or a similar sentence. Is there a reference for the minimum number of necessary reports and/or included patients in order to perform a meta-analysis? P9 - Patient Postoperative Recovery - In addition to the "recovery time of NMB," the authors may wish to also note the incidence of residual neuromuscular block (defined as TOF < 0.90 after tracheal extubation/arrival at PACU).
--

VERSION 2 – AUTHOR RESPONSE

Reviewer: 1

Dr. Alessandro Putzu, Geneva University Hospitals

Comments to the Author:

Thank you for your great work on the manuscript.

I have some minor comments:

1- Abstract. "Data will be synthesized by either fixed-effects or random-effects models according to the I^2 value." This is correct?

Revised. Thank you very much for the reminding of the mistake in the abstract, which was neglect to revise in the first major revision.

2- Trial sequential analysis. I suggest to define a priori a clinical plausible and relevant mean difference to use in the TSA. If it is not the case, it should be reported as a study limitation.

Revised. The primary outcome will be the surgical conditions of the MILS according to the surgeon's perspective. Surgical conditions were evaluated as a surgical rating scale or the percentage of patients with clinically acceptable surgical conditions. It is hard to define a priori a clinical plausible value of relevant mean difference and relative risk increase/decrease during our literature research and clinical experience. So, we reported this deficiency as a study limitation.

3- Risk of bias. I suggest to report in details the items included in the 'other bias' domain (e.g.; financial and non-financial conflict of interest), avoiding general statements such as 'etcetera'.

Revised. Thank you very much for your suggestion. We revised the items of the other bias supplementary additional file 2: Assessment of risk of bias in included studies.

4- Manuscript. I suggest to further improve the clarity and the quality of the English throughout your manuscript.

Our manuscript was revised by the AJE online English Editing and a native English-speaking colleague from our department. But we don't know whether the revised version could meet the requirements for publication. If necessary, we will seek a professional copy-editing service to improve the English quality of our manuscript to the level of publication.

Reviewer: 2

Dr. Sorin Brull, Mayo Clinic

Comments to the Author:

Please review and correct the depth of block definitions listed in the manuscript pages 7-8: deep NMB is defined as TOF count of zero AND (not "or") a PTC ≥ 1 ; intense NMB is defined as TOF count=0 AND (not "or") PTC=0; shallow NMB is defined as TOF count=4 AND (not "or") TOF ratio=0.1-0.4.

Revised. Thank you very much.

Page 8: Types of Outcome Measures. Suggest: "We will perform the meta-analysis only if at least...xxx number of studies have been published in the literature." - or a similar sentence. Is there a reference for the minimum number of necessary reports and/or included patients in order to perform a meta-analysis?

Revised. No reference was related to the minimum number of necessary reports and/or included patients in order to perform a meta-analysis. In most situation, it is not necessary to detail the method of the outcomes reporting such as the minimum number of RCTs during the meta-analysis. As another reviewer mentioned during the first major revision, it is better to report details on outcomes reporting in the protocol (e.g. an outcome will be reported if reported by at least 2 RCTs). So, we revised this part of our protocol.

P9 - Patient Postoperative Recovery - In addition to the "recovery time of NMB," the authors may wish to also note the incidence of residual neuromuscular block (defined as TOF < 0.90 after tracheal extubation/arrival at PACU).

Revised. Thank you very much for your suggestion.